# Your Out-of-Distribution Detection Method is Not Robust!

**Mohammad Azizmalayeri,   Arshia Soltani Moakhar,   Arman Zarei,**
**Reihaneh Zohrabi,   Mohammad Taghi Manzuri,   Mohammad Hossein Rohban**

Department of Computer Engineering
Sharif University of Technology

{m.azizmalayeri, arshia.soltani, arman.zarei, zohrabi, manzuri, rohban}@sharif.edu

## Abstract

Out-of-distribution (OOD) detection has recently gained substantial attention due to the importance of identifying out-of-domain samples in reliability and safety. Although OOD detection methods have advanced by a great deal, they are still susceptible to adversarial examples, which is a violation of their purpose. To mitigate this issue, several defenses have recently been proposed. Nevertheless, these efforts remained ineffective, as their evaluations are based on either small perturbation sizes, or weak attacks. In this work, we re-examine these defenses against an end-to-end PGD attack on in/out data with larger perturbation sizes, e.g. up to commonly used $\epsilon = 8/255$ for the CIFAR-10 dataset. Surprisingly, almost all of these defenses perform worse than a random detection under the adversarial setting. Next, we aim to provide a robust OOD detection method. In an ideal defense, the training should expose the model to almost *all* possible adversarial perturbations, which can be achieved through adversarial training. That is, such training perturbations should based on both in- and out-of-distribution samples. Therefore, unlike OOD detection in the standard setting, access to OOD, as well as in-distribution, samples sounds necessary in the adversarial training setup. These tips lead us to adopt generative OOD detection methods, such as OpenGAN, as a baseline. We subsequently propose the Adversarially Trained Discriminator (ATD), which utilizes a pre-trained robust model to extract robust features, and a generator model to create OOD samples. We noted that, for the sake of training stability, in the adversarial training of the discriminator, one should attack real in-distribution as well as real outliers, but not generated outliers. Using ATD with CIFAR-10 and CIFAR-100 as the in-distribution data, we could significantly outperform all previous methods in the robust AUROC while maintaining high standard AUROC and classification accuracy. The code repository is available at https://github.com/rohban-lab/ATD.

## 1   Introduction

Advances in the deep neural networks have led to their widespread use in real-world applications, such as object detection and image classification [1, 2]. These models have a high degree of generalizability to the extent that they assign an arbitrarily high probability even to the samples not belonging to the training set [3]. This phenomenon causes problems in safety-critical applications like medical diagnosis or autonomous driving that should treat anomalous data differently.

The problem of identifying out-of-distribution data has been widely explored in different categories such as Novelty Detection, Open-Set Recognition and Out-of-Distribution (OOD) detection [4, 5, 6].

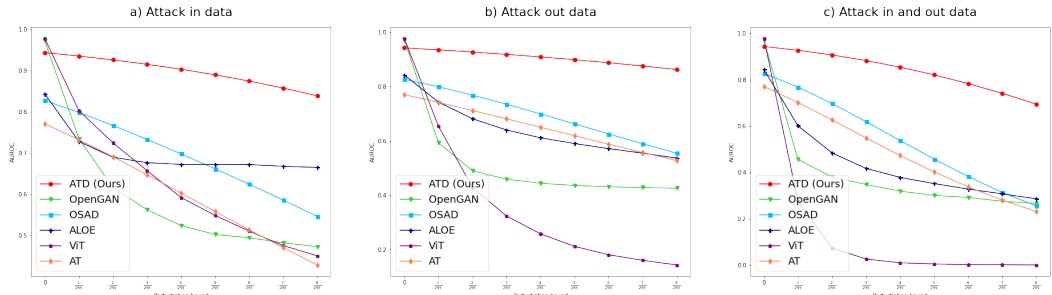

Figure 1: OOD detection scores for several models against the perturbation bound. The perturbations are designed using an end-to-end $\ell_\infty$ PGD attack. CIFAR-10 is used as the in-distribution dataset. a) Only the in-distribution dataset is attacked in the evaluation. b) Only out-distribution datasets (e.g., MNIST, Places, etc.) are attacked in the evaluation. c) Both in- and out-distribution datasets are attacked. ATD (Our method) outperforms others by a significant margin.

These categories are almost similar since they all consider two disjoint sets called the closed (or normal) and open (or anomalous) sets, and the model should discriminate them [7]. So far, outstanding models have been proposed in this research field [8, 9, 10, 11]. Still, these models need to be examined from other aspects, specifically robustness against adversarial perturbations added to the input data.

Deep networks are vulnerable against adversarial examples. This phenomenon was first noticed in the image classification, but it also extends to other domains [12, 13, 14]. Adversarial examples are the inputs that are slightly perturbed with a bounded perturbation (e.g. through bounding the $\ell_p$-norm) to cause a high prediction error in the model. Several defenses have been proposed to make the image classification robust, none of which is as effective as adversarial training and its variants [15, 16, 17, 18]. Similar to other learning domains, OOD detection methods also suffer from the existence of adversarial examples. A small perturbation can cause a sample from the closed set to be classified as anomaly and vice-versa [19, 20].

Robust OOD detection is the connection between these two safety-critical issues. Since a sample does not change semantically under appropriately bounded perturbation in the adversarial attacks, the OOD detector model is expected to be able to label them as open/closed correctly. As a first thought, one might envision solving the issue by using the image classification defenses. Despite advances in these defenses, they have no intuition of the OOD data and not seen them in the training, making them less efficient against end-to-end attacks on the OOD detection method.

Two main approaches are introduced in previous works for the robust OOD detection. The first one uses some OOD data, uniformly sampled from the open set classes and disjoint from the closed set, and employs adversarial training to encourage the model to assign a uniform label, as opposed to one-hot labels, to anomalous samples even under adversarial perturbations [21, 22, 23]. The second approach conducts adversarial training only on the closed set, but encourages the model to learn more semantic features using representation learning techniques such as auto-encoders, self-supervised learning, and denoising methods [24, 25]. Despite the benefits of these methods, they still have fundamental issues such as:

- They did not evaluate their method with an end-to-end attack on the detection method, e.g. they attack the *classification model*, and not the detector, to evaluate the detection method [24, 25]. We show that this is not an effective and strong attack on the detector.

- They consider small perturbation sizes (e.g. $\epsilon = \frac{1}{255}$ in the CIFAR-10 dataset [23], or not attacking the in-distribution [19, 22]) to protect the standard classification accuracy, while we show that images do not change perceptually with larger perturbation sizes (e.g. $\epsilon = \frac{8}{255}$ in the CIFAR-10 dataset). Thus, these methods should be trained and evaluated against larger perturbations.

- The second approach is still unaware of OOD samples during training. Furthermore, the first approach also cannot cover various aspects of OOD data since they can only use a small number of OOD training samples.

In this work, we first evaluate previous defenses as well as standard OOD detection methods against an end-to-end PGD attack [15]. As shown in Fig. 1, almost all of the earlier methods perform worse than random in the binary OOD detection with $\epsilon = \frac{8}{255}$ in CIFAR-10. As a result, the robust

OOD detection method is still far from being solved at present. In spite of this, it is possible to design a robust model by addressing the drawbacks of previous methods. To this end, we propose Adversarialy Trained Discriminator (ATD) that uses a GAN-based method to generate OOD samples and discriminate them from the closed set. An ideal generator naturally deceives the discriminator, and therefore produces adversarial outputs from the open set [26, 27, 28]. This leads to the implicit robustness of the discriminator against adversarial attacks on the open set. In addition, we also conduct adversarial training on the closed set and the tiny real open set (known as outlier exposure) to achieve robustness on them as well. Our results (e.g., Fig. 1) show that ATD outperforms previous methods by a large margin. Therefore, this article has taken a significant step in this area by revealing the vulnerability of all previous methods and providing a solution.

## 2 Background

### 2.1 Out-of-Distribution Detection

**Probabilities:** A simple but effective method for anomaly detection is the Maximum Softmax Probability (MSP) [29, 30]. This method is applied on a $K$-class classifier and returns $\max_{c \in \{1,2,...,k\}} f_c(x)$ as the closed-set membership score of the sample $x$. Currently, it is shown that using the MSP with Vision Transformers (ViT) [31] leads to SOTA results in cross-dataset open set recognition [11, 32]. Also, OpenMax [33] replaces the softmax layer with a layer that calibrates the logits by fitting a class-wise probability model such as the Weibull distribution [34].

**Distances:** The anomaly sample can be detected by its distance to the class conditional distributions. Mahalanobis distance (MD) [35] and Relative MD (RMD) [36] are the main methods in this regard. For an in-distribution with $K$ classes, these methods fit a class conditional Gaussian distribution $\mathcal{N}(\mu_k, \Sigma)$ to the pre-logit features $z$. The mean vector and covariance matrix are calculated as:

$$\mu_k = \frac{1}{N} \sum_{i:y_i=k} z_i, \quad \Sigma = \frac{1}{N} \sum_{k=1}^{K} \sum_{i:y_i=k} (z_i - \mu_k)(z_i - \mu_k)^T, \quad k = 1, 2, ..., K. \qquad (1)$$

In addition, to use RMD, one has to fit a $\mathcal{N}(\mu_0, \Sigma_0)$ to the whole in-distribution. Next, the distances and anomaly score for the input $x'$ with pre-logits $z'$ are computed as:

$$MD_k(z') = (z' - \mu_k)^T \Sigma^{-1} (z' - \mu_k), \quad RMD_k(z') = MD_k(z') - MD_0(z'), \qquad (2)$$

$$score_{MD}(x') = -\min_k \{MD_k(z')\}, \quad score_{RMD}(x') = -\min_k \{RMD_k(z')\}. \qquad (3)$$

**Discriminators:** Previous categories define the score function based on a $K$-way classifier, but one can directly train a binary discriminator for this purpose. Outlier Exposure (OE) [37] exploits some outlier data to learn a binary discriminator for the open set discrimination. GOpenMax [38], OSRCI [39], and Confident Classifier [40] augment the closed set with fake images through a GAN-based generator. Also, OpenGAN [28] selects the best discriminator model with an open validation set during training due to unstable training of GANs.

### 2.2 Adversarial Attacks

For an input $x$ with the ground-truth label $y$, an adversarial example $x^*$ is crafted by adding small noise to $x$ such that the predictor model loss $J(x^*, y)$ is maximized. The $\ell_p$ norm of the adversarial noise should be less than a specified value $\epsilon$, i.e. $\|x - x^*\|_p \leq \epsilon$, to ensure that the image does not change semantically. Fast Gradient Sign Method (FGSM) [13] maximizes the loss function with a single step toward the sign of the gradient of $J(x, y)$ with respect to the $x$:

$$x^* = x + \epsilon \, . \, sign(\nabla_x J(x, y)), \qquad (4)$$

where the noise meets the $\ell_\infty$ norm bound $\epsilon$. Moreover, this method can be conducted iteratively [41] with a smaller step size $\alpha$:

$$x_0^* = x, \quad x_{t+1}^* = x_t^* + \alpha \, . \, sign(\nabla_x J(x_t^*, y)), \qquad (5)$$

where the noise should be projected to the $\ell_\infty$-ball with $\epsilon$ radius at each step, which is called Projected Gradient Descent (PGD) attack [15]. There are a number of other attacks for evaluating robustness of the models [42, 43, 44], but PGD is regarded as a standard and powerful attack.

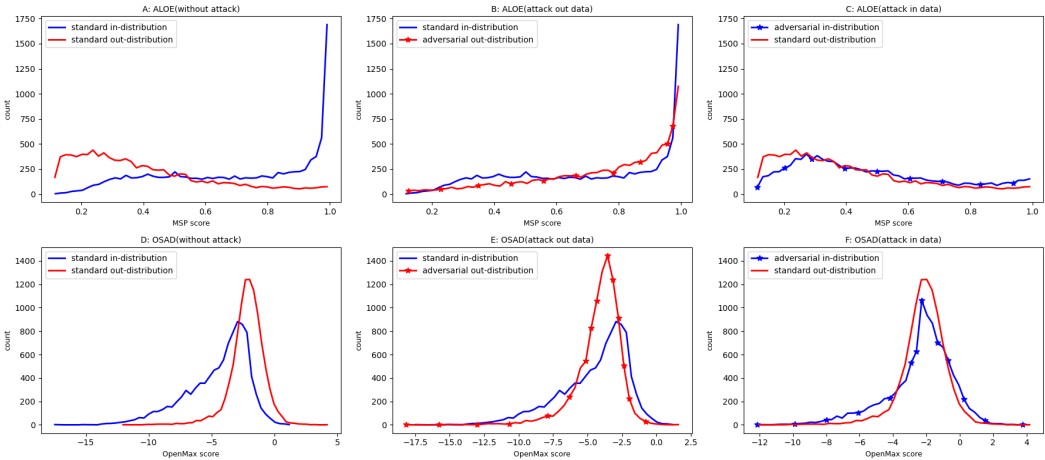

Figure 2: OOD score distribution shift after an end-to-end PGD attack with $\epsilon = \frac{8}{255}$ using CIFAR-10 as the in-distribution set. The first and second row correspond to the ALOE and OSAD methods, respectively. In each row, the left column represents the standard score distributions of in- and out-distributions. In each of the next columns, one of the in or out sets is attacked. The plots show that the standard distribution scores have drastically changed after the attack.

## 2.3 Adversarial Defenses

The most effective defenses for image classification and OOD detection are variants of adversarial training, which are described in the following.

**Adversarial Training (AT)** [15]: Through the use of adversarial examples during training, the model can learn robust features to withstand adversarial perturbations. This method is called AT and can be used to optimize model $f_\theta$ as:

$$\arg\min_\theta \ \mathbb{E}_{(x,y)\in D_{in}} \ \max_{\|x-x^*\|_\infty \leq \epsilon} J(x^*, y; f_\theta), \tag{6}$$

where the inner maximization can be approximated with PGD and the outer minimization with SGD. AT naturally reduces the standard accuracy, i.e. the accuracy on unperturbed samples [45], which can harm the anomaly detection score [11]. Therefore, we also propose Helper-Based Adversarial Training (HAT) [46] as a baseline that achieves a better trade-off between accuracy and robustness.

**Adversarial Learning with inliner and Outlier Exposure (ALOE)** [23]: AT lacks information about the outlier data. Therefore, it is only robust on the closed set and not on outliers. To mitigate this issue, ALOE includes some outliers in the AT similar to OE [37]. Outliers are used with uniform label $\mathcal{U}_K$ and attacked during training to obtain a model that is robust on both in- and out-distribution. The objective function of ALOE is summarized as:

$$\arg\min_\theta \ \mathbb{E}_{(x,y)\in D_{in}} \ \max_{\|x-x^*\|_\infty \leq \epsilon} J(x^*, y; f_\theta) \ + \ \lambda \cdot \mathbb{E}_{x\in D_{out}} \ \max_{\|x-x^*\|_\infty \leq \epsilon} J(x^*, \mathcal{U}_K; f_\theta). \tag{7}$$

An identical objective function is also used in RATIO [21] with a $\ell_2$ perturbation bound. Alternatively, one could consider using clean outliers samples in the training. In addition to ALOE, we evaluate this alternative method as a baseline, and refer to it as Adversarial OE (AOE).

**Open-Set Adversarial Defense (OSAD)** [24, 25]: Instead of including outlier data in the AT, OSAD tries to learn more semantic features in the training. To this end, OSAD adds dual-attentive denoising layers to the model architecture, and combines the AT loss with an auto-encoder loss, whose aim is to reconstruct the clean image from its adversarial version. A self-supervision loss is also added by applying transformations on the input image. They claim that this combination improves OSR robustness, but they did not evaluate the model against an end-to-end attack.

## 3 Method

An OOD detection method should be robust against adversarial perturbations that are added to the closed or open datasets. Despite previous efforts to provide robust OOD detection methods, the

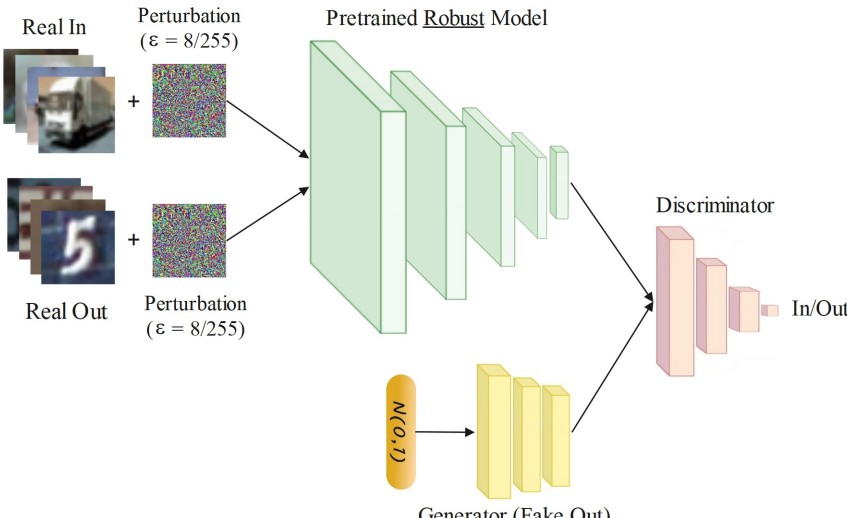

Figure 3: ATD schematic architecture. Generator, discriminator, and robust feature extractor are represented with yellow, pink, and green colors, respectively.

evaluations have yielded a false sense of robustness due to the weak evaluation attacks. In Fig. 2, we have evaluated ALOE and OSAD methods using an end-to-end PGD attack with $\epsilon = \frac{8}{255}$. According to the results, both the in and out detection score distributions remarkably change with this attack. So, these methods are not sufficiently robust. In the following, we aim to provide a more robust solution by addressing the drawbacks of previous works. We then use a toy example to provide more insights into the problem and our solution.

### 3.1 ATD: Adversarially Trained Discriminator

A robust OOD detection method must consider several factors to achieve robustness on both open and closed sets. There has been an extensive study of closed set robustness, which shows that variants of AT are the most effective defenses. Still, pure AT cannot provide robustness in OOD detection for two primary reasons. First, an AT model achieves a lower accuracy due to the trade-off between accuracy and robustness [16, 45]. Note that accuracy plays an important role in the OOD detection [11]. Second, the AT does not consider samples from the open set during training. ALOE tried to mitigate this issue using open datasets in training. However, this strategy works only when the open training data is a close proxy for the OOD data. Unfortunately, this is not true in most cases.

To resolve these issues, we consider OpenGAN as our baseline. Here, the discriminator performs the in/out binary classification. In this setting, a generator crafts OOD images to deceive the discriminator by images similar to the closed set. An ideal generator would cover a broader range of OOD distributions, which is a perfect setting for our problem. Furthermore, as the generator is trained to deceive the discriminator, the generated samples are naturally adversarial examples to the discriminator, which meets the needs in the AT. Hence no adversarial attack is need on such samples. Thus, we only need to perturb the closed set to adversarially train the discriminator. We call this method Adverarially Trained Discriminator (ATD), which can be summarized as:

$$\min_{G} \max_{D} \mathbb{E}_{x \in D_{in}} \log \min_{\|x - x^*\|_{\infty} \leq \epsilon} \mathcal{D}(x^*) + \mathbb{E}_{z \in \mathcal{N}} \log(1 - \mathcal{D}(G(z))), \quad (8)$$

where the $\mathcal{D}$ and $G$ are the discriminator and generator, respectively. Moreover, the inner minimization is approximated with the PGD attack and the outer minimization and maximization with SGD. Using adversarial perturbations to train GANs makes them more robust to adversarial examples, just like other architectures [47, 48]. On the other hand, adding perturbations to the closed set makes it difficult to optimize the discriminator. As a result, the generator would not be trained to cover a broad range since it could easily fool the discriminator in the min-max game for training GANs.

To get closer to the ideal generator that can cover a broad range of OOD distributions, unlike previously proposed robust GANs [47], we make the generator craft features instead of images. It is

evident that generating and discriminating the low-dimensional features is an easier problem than the high-dimensional images, which brings the generator closer to the ideal case. As a result, we also need a model that can extract robust features from the closed set. For this purpose, we use a pre-trained model with the HAT [46] method that achieves a satisfactory accuracy along with robustness on the closed set, considering the trade-off between accuracy and robustness [45]. The last layer, which could be thought of as a linear classifier, is excluded to obtain features instead of logits.

Finally, similar to the OpenGAN and ALOE, we utilize an open dataset in the training to stabilize the training. This data is also attacked during training since it is neither originally adversarial, nor is crafted by the generator. With these in mind, the objective function of the ATD can be reformulated as follows:

$$\min_{G} \max_{\mathcal{D}} \mathbb{E}_{x \in D_{in}} \log \min_{\|x-x^*\|_\infty \leq \epsilon} \mathcal{D}(f_\theta(x^*)) +$$
$$\alpha . \mathbb{E}_{x \in D_{out}} \log(1 - \max_{\|x-x^*\|_\infty \leq \epsilon} \mathcal{D}(f_\theta(x^*))) + (1-\alpha) . \mathbb{E}_{z \in \mathcal{N}} \log(1 - \mathcal{D}(G(z))), \quad (9)$$

where $f_\theta$ is the pre-trained robust feature extractor and $\alpha$ controls the weight of real open dataset in the optimization. Fig. 3 shows a schematic representation of ATD.

## 3.2 Toy Example

Here, we provide an insightful visualization of the ATD using a 2D toy example. To this end, we represent open and closed sets with disjoint distributions in a 2D space. Also, some samples are selected randomly in this space as the generated data. Fig. 4(a) shows the distributions used for this purpose, which the blue, orange, and green samples are closed, open, and generated data, respectively. Next, we train a multi-layer feed-forward neural network as the discriminator to perform the binary classification of in/out on these data in different cases. It should be noted that the open and closed data are fixed during the training, but the generated data is resampled in every epoch to simulate ATD training. Finally, the discriminator output is displayed all over the space with a blue (in) and orange (out) background in each case.

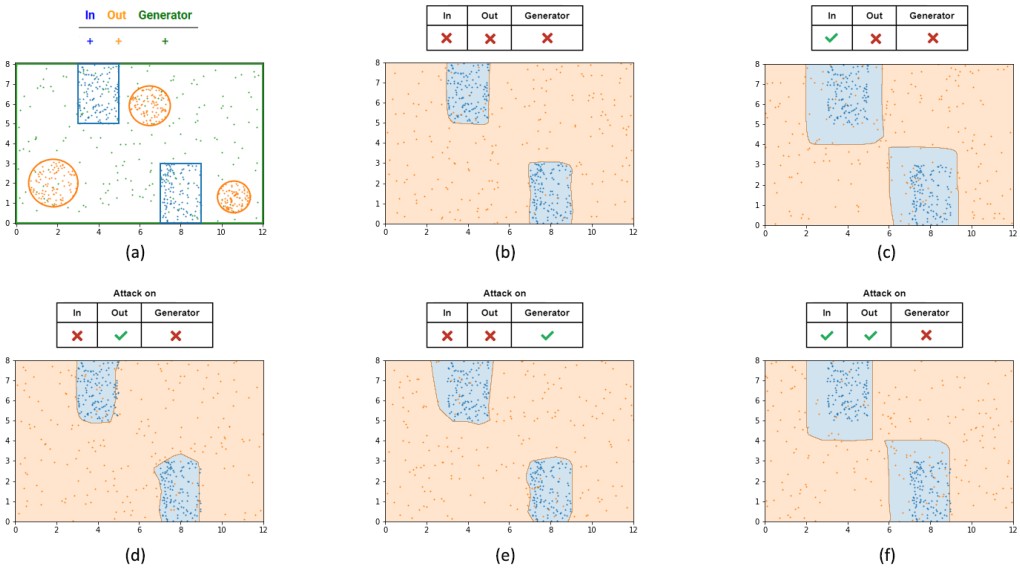

Figure 4: Two-dimensional example to illustrate the effect of ATD method. a) The in- and out-distributions samples that are considered in this example, which are represented with blue and orange colors, respecitvely. In and out samples are fixed, while the generated data colored in green is resampled at each epoch. b to f) Using these distributions, OOD classification has been trained with different attack settings and the classifier output is represented with blue (in) and orange (out) background all over the feature space. Above each plot, green ticks and red crosses indicate which data distributions are attacked.

In our analysis, different conditions are taken into account. First, standard training is conducted on the data without attacking any samples. The results in Fig. 4(b) show that the model has learned the classification decision boundaries very well. In each of the next three cases, one of the closed, open and generated sets is attacked during training with $\epsilon = 1$. Fig. 4(c) shows that attacking the closed set leads to a larger margin in the decision boundaries around them. Having this margin ensures robustness against attacks that seek to misclassify the in-distribution samples. Additionally, note that this margin is smaller in the right side that an open set is present, which shows the impact of using open sets during training. In Fig. 4(d), only the open set is attacked during the training. This has caused a tighter decision boundaries around the closed set where the open set is present. Therefore, using attacked open set in the training can slightly improve the robustness on OOD data, but it can simultaneously reduce the robustness on the closed set. As a result, $\alpha$ in Eq. 9 that controls the optimization weight of open set should be selected carefully. In contrast with the ATD method, we also attack the generated data to see how it affects the result. According to Fig 4(e), this does not cause robustness in the OOD data in contrast to Fig 4(d), which is another reason that ATD do not attack the generated data during training. It is noteworthy that the main cause for this observation may be the instability of the generated samples during the training, which is inevitable and makes the adversarial optimization goal infeasible.

Eventually, we attack both the open and closed sets during training as in ATD. The results in Fig. 4(f) show that the margin around the closed set increases in the directions that an open set is not present to provide robustness on both in and out samples, which can be considered as a mixture of results in Fig. 4(c) and 4(d). Also, this plot demonstrates that the robustness of the open and closed sets may be at odds with each other.

# 4 Experiments

In this section, we perform extensive experiments to evaluate existing OOD detection methods, including the standard and adversarially trained ones, and our ATD method against an end-to-end PGD attack. To this end, we first give details about the setting of the experiments. Next, we compare all the methods, which shows that ATD significantly outperforms the other methods. Toward the end, we conduct some additional experiments to investigate some aspects of our solution.

## 4.1 Setup

**Detection Methods:** Apart from the methods that use discriminators for the detection, other models are evaluated using MSP, OpenMax, MD and RMD as probability and distance based methods.

**In-distribution Datasets:** CIFAR-10 and CIFAR-100 [49] are used as the in-distribution datasets. The images pixel values are normalized to be in the range of 0 to 1.

**Out-of-distribution Datasets:** Following the setting in earlier works [23, 37], we use eight different datasets that are disjoint from the in-distribution sets, including MNIST [50], TinyImageNet [51], Places365 [52], LSUN [53], iSUN [54], Birds [55], Flowers [56], and COIL-100 [57] as the OOD test sets. The results are averaged over these datasets to peform a comprehensive evaluation on different OOD datasets. Also, the SVHN [58] dataset is used as the OOD validation set to select the best discriminator during the training, and Food-101 [59] is used as the open training set.

**Baselines:** Various methods have been considered in our comparisons. The ViT architecture and OpenGAN are used as the SOTA methods in the standard OOD detection. AT and HAT are considered as the effective defenses in the image classification, and AOE, ALOE, and OSAD as effective defenses in the OOD detection. Note that we are the first work that uses HAT for the OOD detection. All the defenses are trained with $\epsilon = \frac{8}{255}$ to have the best results against attack with this perturbation budget.

These baseline methods are trained based on the guidelines in their original work. It should be noted that OpenGAN evaluates the model in the training mode. This causes a kind of information leakage among the batch of images used to predict the OOD scores. Nevertheless, the evaluations of OpenGAN is done according to their guidelines. This problem has been solved in our implementations for ATD method and ATD is evaluated in the test mode.

**ATD Hyperparameters:** A simple DCGAN [60] is used for the generator and discriminator architecture in the ATD. Furthermore, ATD is trained for 20 epochs with $\alpha = 0.5$ using Adam [61] optimizer with learning rate of $1e - 4$. Details of the ATD method is available in section 3.1.

Table 1: OOD detection AUROC under attack with $\epsilon = \frac{8}{255}$ for various methods trained with CIFAR-10 or CIFAR-100 as the closed set. A clean evaluation is one where no attack is made on the data, whereas an in/out evaluation means that the corresponding data is attacked. The best and second-best results are distinguished with bold and underlined text for each column.

| Method | In-Distribution Dataset | | | | | | | |
| --- | --- | --- | --- | --- | --- | --- | --- | --- |
| | CIFAR-10 | | | | CIFAR-100 | | | |
| | Clean | In | Out | In and Out | Clean | In | Out | In and Out |
| OpenGAN-fea | 0.971 | 0.473 | 0.425 | 0.266 | **0.958** | 0.198 | 0.324 | 0.088 |
| OpenGAN-pixel | 0.818 | 0.000 | 0.008 | 0.000 | 0.767 | 0.000 | 0.004 | 0.000 |
| ViT (MSP) | 0.975 | 0.448 | 0.172 | 0.002 | 0.879 | 0.269 | 0.129 | 0.002 |
| ViT (MD) | **0.995** | 0.136 | 0.495 | 0.000 | 0.951 | 0.053 | 0.279 | 0.000 |
| ViT (RMD) | 0.951 | 0.427 | 0.446 | 0.025 | 0.915 | 0.365 | 0.361 | 0.037 |
| ViT (OpenMax) | 0.984 | 0.346 | 0.291 | 0.004 | 0.907 | 0.086 | 0.166 | 0.001 |
| AT (MSP) | 0.735 | 0.462 | 0.442 | 0.174 | 0.603 | 0.324 | 0.250 | 0.085 |
| AT (MD) | 0.771 | 0.429 | 0.527 | 0.232 | 0.649 | 0.278 | 0.357 | 0.108 |
| AT (RMD) | 0.836 | 0.436 | 0.523 | 0.151 | 0.700 | 0.366 | 0.363 | 0.136 |
| AT (OpenMax) | 0.805 | 0.468 | 0.508 | 0.208 | 0.650 | 0.319 | 0.350 | 0.132 |
| HAT (MSP) | 0.770 | 0.560 | 0.548 | 0.325 | 0.612 | 0.393 | 0.335 | 0.176 |
| HAT (MD) | 0.789 | 0.572 | 0.586 | 0.369 | 0.810 | 0.587 | 0.603 | 0.363 |
| HAT (RMD) | 0.878 | 0.602 | 0.606 | 0.258 | 0.730 | 0.443 | 0.416 | 0.191 |
| HAT (OpenMax) | 0.821 | 0.613 | 0.648 | 0.415 | 0.703 | 0.462 | 0.462 | 0.263 |
| OSAD (MSP) | 0.698 | 0.411 | 0.407 | 0.154 | 0.557 | 0.285 | 0.194 | 0.055 |
| OSAD (MD) | 0.626 | 0.375 | 0.432 | 0.231 | 0.615 | 0.368 | 0.416 | 0.216 |
| OSAD (RMD) | 0.776 | 0.421 | 0.456 | 0.123 | 0.680 | 0.369 | 0.353 | 0.140 |
| OSAD (OpenMax) | 0.827 | 0.544 | 0.554 | 0.251 | 0.647 | 0.325 | 0.330 | 0.123 |
| AOE (MSP) | 0.780 | 0.544 | 0.527 | 0.285 | 0.566 | 0.332 | 0.324 | 0.157 |
| AOE (MD) | 0.709 | 0.361 | 0.484 | 0.215 | 0.743 | 0.406 | 0.539 | 0.255 |
| AOE (RMD) | 0.780 | 0.382 | 0.421 | 0.075 | 0.682 | 0.355 | 0.313 | 0.121 |
| AOE (OpenMax) | 0.797 | 0.528 | 0.586 | 0.298 | 0.591 | 0.282 | 0.356 | 0.143 |
| ALOE (MSP) | 0.843 | 0.664 | 0.538 | 0.287 | 0.701 | 0.438 | 0.317 | 0.127 |
| ALOE (MD) | 0.827 | 0.369 | 0.479 | 0.132 | 0.793 | 0.516 | 0.543 | 0.264 |
| ALOE (RMD) | 0.815 | 0.293 | 0.364 | 0.022 | 0.632 | 0.275 | 0.283 | 0.078 |
| ALOE (OpenMax) | 0.868 | 0.584 | 0.606 | 0.261 | 0.731 | 0.399 | 0.389 | 0.125 |
| ATD (Ours) | 0.943 | **0.837** | **0.862** | **0.693** | 0.877 | **0.734** | **0.739** | **0.553** |

**Evaluation Attack:** All the models are evaluated against an end-to-end PGD attack with $\epsilon = \frac{8}{255}$. For the baseline methods, we only use a 10-step attack, but ATD is evaluated with 100 steps to ensure its robustness. We show that this perturbation budget does not change the images semantically and 100 steps is more than enough to ensure the robustness of ATD in Appendix C and E. Also, the attack is performed with a single random restart and the random initialization in the range $(-\epsilon, \epsilon)$. Moreover, the attack step size is selected as $\alpha = 2.5 \times \frac{\epsilon}{N}$.

**Evaluation Metric:** We use AUROC as a well-known classification criterion. The AUROC value is in the range $[0, 1]$, and the closer it is to 1, the better the classifier performance.

## 4.2 Results

**OOD detection under adversarial attack:** To perform a comprehensive study, AUROC is computed in four different settings for each method. First, the standard OOD detection without any attack is conducted (Clean). Next, either in- or out-datasets are attacked (In/Out). Finally, both the in- and out-sets are attacked (In and Out). Note that resistance against the attack to both in- and out-sets is much harder than other cases since the perturbation budget has effectively been doubled. Based on the results in Table 1, ViT+MD and OpenGAN-fea have the best performance in the standard OOD

Table 2: ATD is trained on CIFAR-10 and is evaluated with the transferred attack on in/out data generated by baseline methods (columns). Results show that ATD is sufficiently robust under the transferred black-box attacks.

| Attack from | OpenGAN-fea | ViT (MD) | AT (MD) | HAT (OM) | OSAD (OM) | AOE (OM) | ALOE (OM) |
|---|---|---|---|---|---|---|---|
| In | 0.930 | 0.927 | 0.923 | 0.894 | 0.907 | 0.907 | 0.921 |
| Out | 0.940 | 0.940 | 0.933 | 0.914 | 0.927 | 0.925 | 0.933 |
| In and Out | 0.928 | 0.925 | 0.920 | 0.865 | 0.895 | 0.892 | 0.917 |

Table 3: Ablation study on our method. Other choices for the feature extractor, training method, and generated data attacking are tested on CIFAR-10, but none of them is as effective as the setting used in the ATD.

| Config | | | Attack | | | |
|---|---|---|---|---|---|---|
| Feature Extractor | Adversarial Train | Attack Generator | Clean | In | Out | In and Out |
| Standard Trained | ✓ | ✗ | 0.706 | 0.263 | 0.147 | 0.029 |
| HAT | ✗ | ✗ | **0.947** | 0.741 | 0.720 | 0.511 |
| Not Used | ✓ | ✓ | 0.916 | 0.776 | 0.758 | 0.546 |
| Not Used | ✓ | ✗ | 0.923 | 0.789 | 0.771 | 0.560 |
| HAT | ✓ | ✗ | 0.943 | **0.837** | **0.862** | **0.693** |

detection, but they completely fail under the adversarial setting. Early defenses improved adversarial performance at the cost of a decrease in the clean detection rate. Among them, HAT+OpenMax and HAT+MD are the most effective defenses in CIFAR-10 and CIFAR-100, respectively. Still, they are not as effective as our method ATD, which significantly outperforms them in both datasets while preserving the clean detection performance.

**Black-box evaluation:** To evaluate ATD under the black-box setting, we generate adversarial perturbations by attacking the introduced baselines as the source models and transferring them to the ATD as the target model. This test can also be considered as a sanity check for detecting gradient obfuscation in the model [62]. Since the single-step attack enjoys better transferability than the multi-step ones [41], we use FGSM for attacking the source models. Based on the results in Table 2, ATD is sufficiently robust against all the transferred attacks.

**Ablation study:** ATD uses HAT as the feature extractor and performs adversarial training on open and closed sets to robustify the discriminator. As an ablation study using CIFAR-10 as the in-distribution data, we replace HAT with a standard trained model to check its effectiveness. Also, in another experiment, we replace the discriminator adversarial training with the standard training. The results in Table 3 demonstrate that these settings are not as effective as ATD in achieving a robust detection model. As another ablation, we consider removing the feature extractor and generating images instead of features. Based on the results, the trained discriminator with this method is also not as robust as the ATD. Moreover, we check the effect of attacking the generated images, which leads to a lower AUROC score. This confirms that attacking generated data is not helpful in the training.

**Classification attack instead of end-to-end attack:** OSAD method is evaluated against an attack on the classification, and not the OOD detection, in their original work. Here, we show that this is not a strong attack on the detection method. This is done by checking whether an attack on classification or detection method is effective on the other or not. Results in Table 4 for the OSAD and ATD methods demonstrate that attacking the classification and detection is significantly more effective on their self than the other. Thus, an end-to-end attack is a better basis for the evaluation of the classification or detection defenses.

**Comparison with ATOM:** Our baselines include all the previous methods that consider robustness on both in- and out-of-distribution data to make a fair and comprehensive evaluation of our methods. ATOM method [19] is a defense that considers robustness only on the out-of-distribution data. Therefore, this method would not be robust against attacks on the in-distribution data. In addition, we have compared our results against ALOE, which can be regarded as an extension of ATOM that accounts for both in- and out-of-distribution robustness. Still, a comparison with the ATOM method is performed in the Table 5 using PGD-100 with $\epsilon = \frac{8}{255}$ that supports our arguments.

Table 4: Classification accuracy and OOD detection AUROC under attack to classification and detection methods for OSAD and ATD methods trained on CIFAR-10 dataset. An end-to-end attack is more effective in both classification and detection.

| Evaluation | OSAD | | ATD (Ours) | |
|---|---|---|---|---|
| | Attack Classification | Attack Detection | Attack Classification | Attack Detection |
| Classification Accuracy | 0.419 | 0.777 | 0.622 | 0.847 |
| Detection AUROC | 0.813 | 0.544 | 0.918 | 0.837 |

Table 5: Comparison of AUROC score for OOD detection with ATD and ATOM methods under PGD-100 attack with $\epsilon = \frac{8}{255}$. CIFAR-10 and CIFAR-100 datasets are used as the in-distribution dataset. A clean evaluation is one where no attack is made on the data, whereas an in/out evaluation means that the corresponding data is attacked.

| Method | In-Distribution Dataset | | | | | | | |
|---|---|---|---|---|---|---|---|---|
| | CIFAR-10 | | | | CIFAR-100 | | | |
| | Clean | In | Out | In and Out | Clean | In | Out | In and Out |
| ATOM | **0.983** | 0.156 | 0.447 | 0.067 | **0.925** | 0.089 | 0.728 | 0.085 |
| ATD (Ours) | 0.943 | **0.837** | **0.862** | **0.693** | 0.877 | **0.734** | **0.739** | **0.553** |

## 5 Conclusion

The existing OOD detection methods are far from being robust against strong attacks, contrary to what has been claimed previously for methods such as ALOE and OSAD. To mitigate this issue, we proposed ATD that uses an adversarially trained discriminator to classify in and out samples. Moreover, ATD utilizes HAT to extract robust features from the real samples, and a generator to craft a broad range of OOD data features. This method could significantly outperform earlier methods against an end-to-end PGD attack. Also, it is sufficiently robust against black-box attacks. The primary advantage of ATD is that it preserves the standard OOD detection AUROC along with the robustness, which is needed in real-world situations.

## Broader Impact

Detecting out-of-distribution inputs is a safety issue in many machine learning models. In spite of advances in this area, the existing models are susceptible to adversarial examples as another important safety problem. This work aims at overcoming this issue by detecting OOD inputs even when they have been perturbed by adversarial threat models. We believe that this is important in most machine learning systems with safety-critical concerns. For instance, it is required to diagnose an unseen disease in health care systems or to detect anomalous patterns in financial services even if an adversary has perturbed their inputs. Therefore, this work can benefit a wide range of machine learning researchers. Also, we do not expect our efforts to have any negative consequences.

## Acknowledgments

We thank Mahdi Amiri, Hossein Mirzaei, Zeinab Golgooni, and the anonymous reviewers for their helpful discussions and feedbacks on this work.

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
