# OpenReview forum: "Your Out-of-Distribution Detection Method is Not Robust!"
_NeurIPS.cc/2022/Conference — NeurIPS 2022 Accept_

### Official Review · Reviewer_1FKs · 2022-07-08

**Rating:** 6
**Confidence:** 4
**Soundness:** 2 fair
**Presentation:** 3 good
**Contribution:** 1 poor

**Summary:**

This paper analyzes the robustness of outlier detection, and proposes to adversarially train the discriminator in order to get better robustness. Experiments demonstrate the effectiveness of the method

**Questions:**

The Atom paper is cited as "Similar to other learning domains, OOD detection methods also suffer from the
44 existence of adversarial examples. A small perturbation can cause a sample from the closed set to be
45 classified as anomaly and vise-versa [17, 18]."; so it seems that we only talk about the attack part of that paper, is there a particular reason why we don't consider the defense there? (in particular, that paper consider adversarial training of the discriminator)

**Limitations:**

As I mentioned above, as far as I understand, the idea in this paper is not new and has already been explored in previous work (not one work)

**Strengths And Weaknesses:**

I will start with the most important weakness I saw: The idea of adversarially training the discriminator has already been explored in:

Jiefeng Chen, Yixuan Li, Xi Wu, Yingyu Liang, and Somesh Jha. Atom: Robustifying out-of-distribution
337 detection using outlier mining. In Joint European Conference on Machine Learning and Knowledge
338 Discovery in Databases, pages 430–445. Springer, 2021.

And also the recent ICML paper (pushing this even further):

https://arxiv.org/abs/2206.13687 (ICML 2022 (Long Talk))

This paper cited the first paper, but never did comparison in experiments (not to mention the second paper), which I find to be a main weakness.

On the strengths side, this paper does a good job in explaining the ideas and the results are good

---

> ### Author Response · Authors · 2022-08-01
> **Response to Reviewer 1FKs**
>
> Thank you for your review and useful comments. We have made a revision based on the reviewers' concerns, and specific comments are answered below.
>
> 1. We do not claim that our work is the first work that explores robust OOD detection. Previous works have tried to use adversarial training for robust OOD detection, but our experiments (e.g., Fig. 1, Fig. 2, and Table 1)  show that existing models are not sufficiently robust contrary to their claims. To resolve this problem, we provide a new method that outperforms all previous works with a significant margin.
>
> 2. In our experiments, we compare our method with all the previous methods that consider robustness on both in- and out-of-distribution data to make a fair and comprehensive evaluation of our methods. ATOM [Chen et al. 2021] only considers robustness on the out-of-distribution data. Therefore, this method would not be robust against attacks on the in-distribution data. In addition, we have compared our results against ALOE, which can be regarded as an extension of ATOM that accounts for both in- and out-of-distribution robustness.
> Yet, a comparison with the ATOM method is performed in the Table below using PGD-100 with $\epsilon = \frac{8}{255}$ that supports our arguments. A clean evaluation is one where no attack is made on the data, whereas an in/out evaluation means that the corresponding data is attacked.
>
> | in-Distribution | Method | Clean | In | Out | In and Out |
> |-----------------|:---------:|:-----:|:-----:|:-----:|:----------:|
> | CIFAR-10 | ATOM | 0.983 | 0.156 | 0.447 | 0.067 |
> | CIFAR-10 | ATD(Ours) | 0.985 | 0.942 | 0.717 | 0.485 |
> | CIFAR-100 | ATOM | 0.925 | 0.089 | 0.728 | 0.085 |
> | CIFAR-100 | ATD(Ours) | 0.941 | 0.854 | 0.667 | 0.476 |
>
> 3. The ICML paper you mentioned is arxived a month after our submission! Moreover, this paper only tries to select better outliers during standard training, which would not be as effective as adversarial training against adversarial perturbations.
>
>
>
>
> References:
>
> Jiefeng Chen, Yixuan Li, Xi Wu, Yingyu Liang, and Somesh Jha. Atom: Robustifying out-of-distribution 337 detection using outlier mining. In Joint European Conference on Machine Learning and Knowledge 338 Discovery in Databases, pages 430–445. Springer, 2021.

---

### Official Review · Reviewer_47ku · 2022-07-11

**Rating:** 6
**Confidence:** 4
**Soundness:** 3 good
**Presentation:** 3 good
**Contribution:** 2 fair

**Summary:**

he authors of this paper claim to be showing that the current state-of-the art models for the adversarially robust detection of out-of-distribution samples are not effective when evaluated on sufficiently strong threat models. They then go on to propose a method that they claim remedies this issue by incorporating GAN training and manages to simultaneously achieve high clean OOD detection performance, adversarially robust OOD detection and high accuracy.

**Questions:**

Minor comments:
- They refer to OOD detectors that use the softmax score as "likelihood"-based. Likelihood-based methods generally refer to models that fit the density of the data distribution $p(x)$.
- The authors forgot to delete the instructions part of the checklist.

**Limitations:**

I don't think the authors adequately discuss limitations of their approach. For example, they do not acknowledge that similar to AT, their model's accuracy is significantly reduced. They should also openly discuss how computationally expensive their method is.
I agree with the authors' self-assessment in the checklist that their work is unlikely to have negative ethical implications.

**Strengths And Weaknesses:**

The paper generally tackles an interesting problem of adversarially attacking out-of-distribution samples and they choose a fairly standard threat model. Unfortunately, the novelty of their approach is quite limited because they omit a lot of prior benchmarks. For example, they write "The first [approach] uses some OOD data, uniformly sampled from the open set classes and disjoint from the closed set, and employs adversarial training to encourage the model to assign a uniform label, as opposed to one-hot labels, to anomalous samples even under adversarial perturbations [19]." This describes the method ACET [Hein19], but instead they cite ATOM [19] (which uses an out-class) and which they nonetheless don't compare to for some reason. Both methods should be used as baselines. They furthermore use ALOE as a baseline which according to Eq. 7 is identical to the work RATIO [Augustin20] which should at the very least be cited in this context. On top of this, there even exist several works that tackle the issue of unreliable evaluations of adversarial OOD robustness by providing provable guarantees [Bitterwolf20, Meinke21]. Clearly, these are relevant baselines as well and should at the very least be cited. Additionally, even the idea of using GANs in order to produce adversarial OOD samples has previously been proposed in the well-known paper by [Lee18].

The authors point out that prior works supposedly only studied "small perturbation sizes (e.g. $\epsilon=\frac{1}{255}$ in the CIFAR-10 dataset)". It probably makes sense to explicitly cite exactly which papers they are referring to here. A similar comment applies to a statement such as "The existing OOD detection methods are far from being robust against strong attacks, contrary to what has been claimed previously."

The paper uses 8 out-distributions for each in-distribution which is good. Unfortunately, they do not list the individual results for each out-distribution which makes it very hard to check or compare their results. These need to be listed individually in the appendix. Also, some of the out-distributions clearly overlap with the classes of the in-distributions, for example birds and the bird class from CIFAR10. This needs to be discussed somewhere in the paper, or alternatively different out-distributions should be used.

The authors' method seems to work without relying on a difficult training out-distribution. While they do use SVHN for this, it is quite impressive that their training apparently generalizes to out-distributions that are very different from this.

The authors should provide slightly more details on their evaluation attack, i.e. initialization, restarts and step size, potentially in the appendix.

Overall, however, the paper's results seem encouraging. If their model does not get broken by a stronger attack, the fact that they attain high adversarially robust OOD detection performance and clean OOD detection performance at the same time is significant.
They attempt to account for potential gradient obfuscation in the paper via transfer attacks which is good.
They also provide both their code as well as pre-trained models which is also good.

The authors' claim that their method achieves "high accuracy" seems misleading. The backbone feature extractor that they use has a clean accuracy of 91.47% on CIFAR10, which of course is far from "high" compared to SOTA CIFAR10 models. As far as I can tell, this is not pointed out anywhere in the paper.

I believe the paper cannot be accepted in its current form, because it a) does not individually list the performance with respect to different out-distributions and b) does not adequately discuss how its method relates to and compares to prior work that is substantially similar. If these points are adequately addressed I would be willing to raise my score, because the results are indeed promising.

[Hein19] Why relu networks yield high-confidence predictions far away from the training data and how to mitigate the problem, Hein, Andriushchenko, Bitterwolf, @CVPR'19

[Augustin20] Adversarial robustness on in-and out-distribution improves explainability, Augustin, Meinke, Hein @ECCV'20

[Bitterwolf20] Certifiably adversarially robust detection of out-of-distribution data, Bitterwolf, Meinke, Hein @NeurIPS'20

[Lee18] Training confidence-calibrated classifiers for detecting out-of-distribution samples, Lee, Lee, Lee, Shin, @ICLR'18

[Meinke21] Provably Robust Detection of Out-of-distribution Data (almost) for free, Meinke, Bitterwolf, Hein @UDL workshop, ICML'21

---

> ### Author Response · Authors · 2022-08-01
> **Response to Reviewer 47ku (part 1)**
>
> Thank you for your thoughtful and detailed review. We have made a revision based on the reviewers' concerns. Please see our updated version and the answers to the specific comments below.
>
> 1. We have updated the submission to reference the mentioned papers as follows.
>
> >1.1 "The first [approach] uses some OOD data, uniformly sampled from the open set classes and disjoint from the closed set, and employs adversarial training to encourage the model to assign a uniform label, as opposed to one-hot labels, to anomalous samples even under adversarial perturbations [19]." This describes the method ACET [Hein19], but instead they cite ATOM.
> >> The previous citation was ALOE and not ATOM. ACET and RATIO are also added as new references. (line 55)
>
> >1.2 Eq. 7 is identical to the work RATIO [Augustin20] which should at the very least be cited in this context.
> >> This is added after Eq. 7. (line 129)
>
> >1.3 there even exist several works that tackle the issue of unreliable evaluations of adversarial OOD robustness by providing provable guarantees [Bitterwolf20, Meinke21].
> >> Appendix I is added about the certified defenses. (line 564)
>
> >1.4 The idea of using GANs in order to produce adversarial OOD samples has previously been proposed in the well-known paper by [Lee18].
> >> This was discussed in section 2.1 (discriminators). The mentioned reference is also added. (line 100)
>
> >1.5 The authors point out that prior works supposedly only studied "small perturbation sizes (e.g. in the CIFAR-10 dataset)". It probably makes sense to explicitly cite exactly which papers they are referring to here.
> >> The references are added. ALOE uses $\epsilon = \frac{1}{255}$, and ACET and ATOM do not perturb the in-distribution in training. (line 63)
>
> >1.6 A similar comment applies to a statement such as "The existing OOD detection methods are far from being robust against strong attacks, contrary to what has been claimed previously."
> >> The references are added. For instance, based on Table 4, using OSAD paper setting, one would get AUROC=0.813 on CIFAR-10 with our OOD sets, while we could reduce it to 0.543 with an end-to-end PGD-10 attack.  (line 283)

---

> > ### Author Response · Authors · 2022-08-01
> > **Response to Reviewer 47ku (part 2)**
> >
> > 2. Other comments are addressed below.
> >
> > >2.1 They nonetheless don't compare to [ATOM and ACET] for some reason.
> > >> ACET and ATOM only consider robustness on the out-of-distribution data. Therefore, these methods would not be robust against attacks on the in-distribution data. In addition, we have compared our results against ALOE, which can be regarded as an extension of ATOM that accounts for both in- and out-of-distribution robustness.
> > Still, a comparison with the ATOM method is performed in the Table below using PGD-100 with $\epsilon = \frac{8}{255}$ that supports our arguments. A clean evaluation is one where no attack is made on the data, whereas an in/out evaluation means that the corresponding data is attacked. (Added to Appendix H)
> >
> > | in-Distribution | Method | Clean | In | Out | In and Out |
> > |-----------------|:---------:|:-----:|:-----:|:-----:|:----------:|
> > | CIFAR-10 | ATOM | 0.983 | 0.156 | 0.447 | 0.067 |
> > | CIFAR-10 | ATD(Ours) | 0.985 | 0.942 | 0.717 | 0.485 |
> > | CIFAR-100 | ATOM | 0.925 | 0.089 | 0.728 | 0.085 |
> > | CIFAR-100 | ATD(Ours) | 0.941 | 0.854 | 0.667 | 0.476 |
> >
> > >2.2 They do not list the individual results for each out-distribution which makes it very hard to check or compare their results. These need to be listed individually in the appendix.
> > >> The individual results for each out-distribution are added in Appendix J. An example on CIFAR-10 is provided below.
> >
> > | Method | Attack Distribution | MNIST | TiImgNet | Places | LSUN | iSUN | Birds | Flowers | COIL | Average |
> > |:--------------|:-------------------:|:-----:|:------------------------------------------------:|:------------------------------------------:|:------------------------------------:|:------------------------------:|:-----------------------:|:------------------:|:------------:|:-------:|
> > | ALOE (MSP) | Clean | 0.746 | 0.821 | 0.851 | 0.987 | 0.983 | 0.799 | 0.790 | 0.768 | 0.843 |
> > | | In | 0.463 | 0.609 | 0.661 | 0.936 | 0.922 | 0.591 | 0.585 | 0.541 | 0.664 |
> > | | Out | 0.521 | 0.470 | 0.478 | 0.757 | 0.743 | 0.463 | 0.437 | 0.434 | 0.538 |
> > | | In and Out | 0.227 | 0.216 | 0.228 | 0.516 | 0.504 | 0.218 | 0.196 | 0.193 | 0.287 |
> > | ATD (Ours) | Clean | 0.998 | 0.936 | 0.969 | 0.996 | 0.993 | 0.997 | 0.997 | 0.997 | 0.985 |
> > | | In | 0.995 | 0.755 | 0.874 | 0.978 | 0.962 | 0.991 | 0.990 | 0.992 | 0.942 |
> > | | Out | 0.988 | 0.580 | 0.547 | 0.626 | 0.621 | 0.825 | 0.711 | 0.835 | 0.717 |
> > | | In and Out | 0.966 | 0.279 | 0.258 | 0.326 | 0.321 | 0.594 | 0.448 | 0.691 | 0.485 |
> >
> > >2.3 Some of the out-distributions clearly overlap with the classes of the in-distributions, for example birds and the bird class from CIFAR10. This needs to be discussed somewhere in the paper.
> > >> These cases can be regarded as the near-OOD data that the model
> > is expected to detect them as well as the other OODs [Fort et al. 2021]. Therefore, averaging across all these datasets helps to evaluate the models even on the near-OOD data. (Added in line 588)
> >
> > >2.4 The authors should provide slightly more details on their evaluation attack, i.e. initialization, restarts and step size, potentially in the appendix.
> > >> The details of the evaluation attack are added at the beginning of Appendix F.
> >
> > >2.5 If their model does not get broken by a stronger attack, ...
> > >> The model is evaluated against AutoAttack in Appendix F, and the results are still promising.
> >
> > >2.6 The authors' claim that their method achieves "high accuracy" seems misleading. The backbone feature extractor that they use has a clean accuracy of 91.47% on CIFAR10, which of course is far from "high" compared to SOTA CIFAR10 models.
> > >> 91.47% on CIFAR10 is a high clean accuracy with respect to the trade-off between accuracy and robustness. In other words, the accuracy of the backbone feature extractor is higher than other robust models (e.g. pure adversarial training with early stopping reaches 81.66%). (line 175)
> >
> > >2.7 They should also openly discuss how computationally expensive their method is.
> > >> Using the RTX 2060 super and the setting mentioned in the setup section, our model training and evaluation last for 9.5 and 2.5 hours, respectively. The evaluation time of the other methods is similar when using the same setting and MSP detector, but it takes five times longer with distance-based detectors.
> >
> >
> >
> >
> >
> > References:
> >
> > Fort, S., Ren, J. and Lakshminarayanan, B., 2021. Exploring the limits of out-of-distribution detection. Advances in Neural Information Processing Systems, 34, pp.7068-7081.

---

> > > ### Comment · Reviewer_47ku · 2022-08-08
> > > **Acknowledgement of the Rebuttal**
> > >
> > > I want to thank the reviewers for their thorough response to each of my questions and points of criticism. I appreciate the changes that the authors made to the main paper and the appendix (especially the inclusion of Appendix J).
> > >
> > > I will raise my score and recommend this paper for acceptance.
> > >
> > > I would nonetheless like to suggest that the computational resources needed for training (which the authors helpfully stated in their rebuttal) be included in the appendix.
> > >
> > > Also, I do not think that it is the case that distributions with overlapping classes should necessarily be regarded as near-OOD. However, this is not a clean-cut question so I think the authors' approach as described in Appendix J is nonetheless valid. The only thing I would suggest is, that given that near-OOD matters to the authors, they should include CIFAR100 as OOD for CIFAR10 and vice-versa.

---

> > > > ### Author Response · Authors · 2022-08-08
> > > > **Response**
> > > >
> > > > Thank you for the constructive comments and suggestions and for updating the score.
> > > >
> > > > We will add the computational resources needed for training and evaluation in the appendix. We will also add CIFAR-10 vs. CIFAR-100 as the near-OOD test.
> > > >
> > > > Update: Added. See comment above.

---

> > > > ### Author Response · Authors · 2022-08-09
> > > > **Response**
> > > >
> > > > The computational cost for training and evaluation is added in Appendix K.
> > > >
> > > > The results of OOD detection on CIFAR-100 vs CIFAR-10 are added to Tables 10 and 11, which are also provided below. For the baseline methods which are evaluated with different detection methods such as MSP, MD, RMD, and Openmax, only the one with the best average across all the ''Clean'', ''In'', ''Out'', ''In and Out'' cases is chosen.
> > > >
> > > > ID: CIFAR-10, OOD: CIFAR-100
> > > >
> > > > |    Method     | Clean  |   In   |  Out   | In  and  Out  |
> > > > |:-------------:|:------:|:------:|:------:|:-------------:|
> > > > |   ATD(Ours)   | 0.950  | 0.797  | 0.583  |     0.277     |
> > > > |  OpenGAN-fea  | 0.950  | 0.316  | 0.296  |     0.101     |
> > > > |  AT(OpenMax)  | 0.796  | 0.444  | 0.439  |     0.160     |
> > > > |   ALOE(MSP)   | 0.788  | 0.458  | 0.476  |     0.170     |
> > > > | AOE(OpenMax)  | 0.782  | 0.492  | 0.530  |     0.252     |
> > > > | HAT(OpenMax)  | 0.805  | 0.579  | 0.606  |     0.368     |
> > > > | OSAD(OpenMax) | 0.799  | 0.474  | 0.484  |     0.181     |
> > > > |   VIT(RMD)    | 0.973  | 0.443  | 0.364  |     0.017     |
> > > >
> > > >
> > > > ID: CIFAR-100, OOD: CIFAR-10
> > > >
> > > > |   Method    | Clean  |   In   |  Out   | In  and  Out  |
> > > > |:-----------:|:------:|:------:|:------:|:-------------:|
> > > > |   ATD(Ours) | 0.911  | 0.761  | 0.490  |     0.263     |
> > > > | OpenGAN-fea | 0.929  | 0.299  | 0.276  |     0.091     |
> > > > |   AT(RMD)   | 0.675  | 0.320  | 0.295  |     0.107     |
> > > > |  ALOE(MD)   | 0.436  | 0.131  | 0.168  |     0.030     |
> > > > |   AOE(MD)   | 0.443  | 0.149  | 0.236  |     0.061     |
> > > > |   HAT(MD)   | 0.540  | 0.268  | 0.291  |     0.108     |
> > > > |  OSAD(MD)   | 0.503  | 0.261  | 0.277  |     0.103     |
> > > > |  VIT(RMD)   | 0.948  | 0.386  | 0.485  |     0.058     |

---

### Official Review · Reviewer_enTM · 2022-07-11

**Rating:** 7
**Confidence:** 4
**Soundness:** 3 good
**Presentation:** 4 excellent
**Contribution:** 3 good

**Summary:**

This paper uncovers the vulnerability of the existing OOD detection methods to adversarial inputs. It then goes on to introduce a robust OOD detection method called Adversarially Trained Discriminator (ADT). ADT combines adversarial training with generative models to improve the detection robustness. In particular, the GAN model in this approach allows generating OOD samples beyond the available open sets during training and therefore boosts the performance of the detection method as a whole. The idea is simple and makes use of recent related work including HAT. While none of the components of the proposed model are novel, the particular system is a novel combination of these components. Results show substantial improvement in robustness beyond other baselines considered.


POST-REBUTTAL UPDATE:
I thank the authors for responding to my comments and appreciate the new experiments in support of their arguments. I am increasing my score by 1 to recommend acceptance.

**Questions:**

The text was clear and I do not have any additional questions.

**Limitations:**

While CIFAR10 and CIFAR100 are good starting points, many of the past methods have been shown to not scale well to larger, more sophisticated datasets. It would great to see some experiments on how this model would scale to other datasets like tinyimagenet.

**Strengths And Weaknesses:**

**strengths**
- The method is simple and intuitively makes sense.
- The text is well written and a helpful toy example is included to help the reader better understand the effect of the different components of the model.
- The results section involves comparison across many different baselines.

**weaknesses**
Adversarial robustness has been a constantly evolving field in the past years and the frequent breakage of SOTA defenses by new adversarial attacks has made everyone working in this field very cautious about new proposed defenses. Much of recent work in the AT field points to the insufficiency of PGD as a reliable attack method and many defense methods in recent years have successfully "fooled" this attack without providing a "robust" form of robustness. AutoAttack has become a more reliable substitute to PGD and have proven to be much more difficult to be fooled. In the light of this, I would like to see how the proposed method stands against this stronger attack to be more sure about the reliability of the present results.

typo: line 45 vise-versa

Fig2. OpenMax score is not defined which makes it hard to judge what this figure is telling.

---

> ### Author Response · Authors · 2022-08-01
> **Response to Reviewer enTM**
>
> We thank the reviewer for the positive and constructive feedback. We have made a revision based on the reviewers' concerns, and specific comments are answered below.
>
> 1. AutoAttack [Croce et al. 2020] propose to use an ensemble of four diverse attacks to reliably evaluate the robustness. These attacks are run sequentially on an input batch, and the one that causes the most loss is used for perturbing the input. To ensure the reliability of our results, we have evaluated our method against such a strong attack in addition to PGD-100 in the table below. A clean evaluation is one where no attack is made on the data, whereas an in/out evaluation means that the corresponding data is attacked. The results are still promising against AutoAttack.
>
> | in-Distribution | Attack | Clean | In | Out | In and Out |
> |-----------------|:-----------:|:-----:|:-----:|:-----:|:----------:|
> | CIFAR-10 | PGD | 0.985 | 0.942 | 0.717 | 0.485 |
> | CIFAR-10 | Auto Attack | 0.985 | 0.941 | 0.710 | 0.476 |
> | CIFAR-100 | PGD | 0.941 | 0.854 | 0.667 | 0.476 |
> | CIFAR-100 | Auto Attack | 0.941 | 0.852 | 0.663 | 0.469 |
>
>
> 2. Adversarial robustness on larger and sophisticated datasets is still a challenging issue, even in the closed set classification. For instance, the adversarial accuracy on the closed set in the TinyImageNet dataset ($\epsilon = \frac{8}{255}$) is less than 20% [Rade et al. 2021]. On the other hand, the closed set accuracy significantly affects the OOD detection rate [Vase et al. 2021]. As a result, adversarially robust OOD detection on large-scale datasets would be much harder since the robust closed set classification itself is still an open challenging issue.
> Still, we run an experiment on TinyImageNet as the in-distribution dataset that compares our method (using AT as the feature extractor) with ViT, ALOE, and AT methods (all four detectors are used, and the best one is selected). ViT, ALOE, and AT are evaluated against PGD-10 with $\epsilon = \frac{4}{255}$, while PGD-100 with $\epsilon = \frac{4}{255}$ is used for ATD to ensure its robustness. Also, TinyImageNet is excluded from the OOD datasets. The results are listed below that demonstrate our method’s robustness.
>
> | Method | Clean | In | Out | In and Out |
> |:-----------:|:---------:|:---------:|:---------:|:----------:|
> | ViT (MD) | 0.922 | 0.165 | 0.242 | 0.001 |
> |AT (OpenMax) | 0.546 | 0.250 | 0.298 | 0.114 |
> | ALOE (MD) | 0.709 | 0.013 | 0.368 | 0.003 |
> | ATD (Ours) | **0.965** | **0.769** | **0.702** | **0.384** |
>
>
> 3. In Fig. 2, we aimed to show that the distribution of OOD scores significantly changes under attack such that the in- and out-of-distribution samples are not separable anymore within ALOE and OSAD methods. Our numerical results in Table 1 also confirm this. Still, a detailed description of the OpenMax score is available in section 3 of the OSAD paper [Shao et al. 2020].
>
>
>
>
>
> References:
>
> Croce, F. and Hein, M., 2020, November. Reliable evaluation of adversarial robustness with an ensemble of diverse parameter-free attacks. In International conference on machine learning (pp. 2206-2216). PMLR.
>
> Rade, R. and Moosavi-Dezfooli, S.M., 2021, September. Reducing Excessive Margin to Achieve a Better Accuracy vs. Robustness Trade-off. In International Conference on Learning Representations.
>
> Vaze, S., Han, K., Vedaldi, A. and Zisserman, A., 2021. Open-set recognition: A good closed-set classifier is all you need. In International Conference on Learning Representations.
>
> Shao, R., Perera, P., Yuen, P.C. and Patel, V.M., 2020, August. Open-set adversarial defense. In European Conference on Computer Vision (pp. 682-698). Springer, Cham.

---

### Official Review · Reviewer_2bsy · 2022-07-26

**Rating:** 3
**Confidence:** 4
**Soundness:** 2 fair
**Presentation:** 3 good
**Contribution:** 2 fair

**Summary:**

This paper presents an OOD detection method that combines robust feature extraction, OOD feature generation and adversarial training. This Adversarially Trained Discriminator aims to provide a robust OOD detection result that can adapt to larger perturbation sizes. The experiment is conducted on different detector-extractor pairs with respect to applying attack on different data, and the result shows the proposed method consistently outperforms the SoTA methods.


**Questions:**

> Since It’s a key point for the performance improvement, there should be more detailed description about how to use HAT to extract features.

> I don't see any analysis for the different OOD types. Ad the authors stated in Sec. 5, “Moreover, ATD utilizes HAT to extract robust features from the real samples, and a generator to craft a broad range of OOD data features”. However, there is no experiment setup relating to different OOD types, which may not prove ATD being robust enough.

**Limitations:**

As the authors have mentioned, this method significantly outperforms other SOTA OOD detection methods when end-to-end PGD attack is applied. However, further justification should be applied to evaluate the effectiveness/contribution of each detector-extractor pair on different OoD types.


**Strengths And Weaknesses:**

++ This paper specifically pays attention to generating OOD samples and applying Adversarial Training on it to improve the OOD detection performance.

++The paper is well written and easy to follow.

-- Table 3 is not self-contained. It does not mention which dataset are used.

-- In the setup of the experiments, the In-Distribution datasets only contain CIFAR-10 and CIFAR-100 but do not set any combinations with the other data sets like these shown in Out-of-Distribution datasets. Experiments with the different combinations of the ID / OOD dataset could be more convincing.

---

> ### Author Response · Authors · 2022-08-01
> **Response to Reviewer 2bsy**
>
> Thank you for your review and useful comments. We have made a revision based on the reviewers' concerns, and specific comments are answered below.
>
> 1. In Table 3, the in-distribution dataset is CIFAR-10. The other settings are as mentioned in section 4.1 (setup).
>
> 2. The pre-trained PreactResNet model with the HAT method is used as the feature extractor by removing the linear classifier layer (last layer) from the model. This lets us have a 512-dimensional feature vector for each sample.
>
> 3. The in-distribution (two datasets) and out-distribution (eight datasets) are selected following the settings in prior works [Chen et al. 2021, Chen et al. 2022, Katz-Samuels2022], but we agree that different combinations of the ID/OOD datasets should also be investigated separately. Hence, we added our detailed results on different combinations of ID/OOD in appendix J.
>
> 4. A new experiment is conducted using the TinyImageNet as the closed set. In this experiment, ATD (using AT as the feature extractor) is compared against ViT, ALOE, and AT methods (all four detectors are used, and the best one is selected). ViT, ALOE, and AT are evaluated against PGD-10 with $\epsilon = \frac{4}{255}$, while PGD-100 with $\epsilon = \frac{4}{255}$ is used for ATD to ensure its robustness. Also, TinyImageNet is excluded from the OOD datasets. The results are listed below that demonstrate our method's robustness. A clean evaluation is one where no attack is made on the data, whereas an in/out evaluation means that the corresponding data is attacked. Note that adversarially robust OOD detection on such a large dataset is much harder since the robust closed set classification itself is not yet solved on these datasets.
>
> | Method | Clean | In | Out | In and Out |
> |:-----------:|:---------:|:---------:|:---------:|:----------:|
> | ViT (MD) | 0.922 | 0.165 | 0.242 | 0.001 |
> |AT (OpenMax) | 0.546 | 0.250 | 0.298 | 0.114 |
> | ALOE (MD) | 0.709 | 0.013 | 0.368 | 0.003 |
> | ATD (Ours) | **0.965** | **0.769** | **0.702** | **0.384** |
>
>
>
>
> References:
>
> Jiefeng Chen, Yixuan Li, Xi Wu, Yingyu Liang, and Somesh Jha. Robust Out-of-distribution Detection for Neural Networks. In The AAAI-22 Workshop on Adversarial Machine Learning and Beyond.
>
> Jiefeng Chen, Yixuan Li, Xi Wu, Yingyu Liang, and Somesh Jha. Atom: Robustifying out-of-distribution detection using outlier mining. In Joint European Conference on Machine Learning and Knowledge Discovery in Databases, pages 430–445. Springer, 2021.
>
> Katz-Samuels, J., Nakhleh, J.B., Nowak, R. and Li, Y., 2022, June. Training ood detectors in their natural habitats. In International Conference on Machine Learning (pp. 10848-10865). PMLR.

---

### Public Comment · ~Márk_Jelasity1 · 2023-01-29
**previous work**

Dear authors, this is a nice paper, congratulations. I just wanted to let you know, that we published a paper in 2021 with very similar goals  (but not using GANs) that include end-to-end robustness evaluation and a principled, combined in- and out-of-distribution adversarial training methodology as well. The paper is called "Robust Classification Combined with Robust out-of-Distribution Detection: An Empirical Analysis".  http://publicatio.bibl.u-szeged.hu/22470/1/ijcnn21.pdf You might want to consider mentioning us in future works. Thanks.

---

### Public Comment · ~John_Kirk1 · 2023-06-20
**The claim in the abstract that OOD methods violate their purpose because are susceptible to adversarial attacks is not entirely true**

The authors state that OOD methods are still susceptible to adversarial examples, **which is a violation of their purpose**. Indeed, OOD methods are susceptible to adversarial examples, that is not something new (e.g. [here](https://arxiv.org/abs/2009.01798) in pages 13-15, and numerous other works).

What I would like to bring to authors attention is the fact that this is not a violation of their purpose, and here's why. There's a clear separation in the literature between natural and non-natural distribution shifts. Adversarial examples fall under the category of non-natural shifts. But, that's not an issue since we are only playing with semantics so far.

In natural (a.k.a more realistic) datasets the distribution between in-distribution vs out-of-distribution (adversarially perturbed set) does not change that much irrespective of whether the adversarial attack is 1/255 or 8/255, that is why most OOD method will fail.

What is missing from the paper are some (normality) plots showing (like in [here](https://arxiv.org/abs/1801.02257) Figure 4, page 5) actually how different are the distributions between in-distribution vs out-of-distribution (adversrially perturbed set) datasets with both $\epsilon$ = 1/255 and 8/255

---

> ### Public Comment · ~Mohammad_Azizmalayeri1 · 2023-06-25
> **Clarification**
>
> Thanks for your comment. I'm not sure if I completely understood what you meant, but here are my clarifications.
>
> - Our goal is to detect images with different semantics from in distribution as OOD even under adversarial perturbations. In other words, the in and out data in our experiments are semantically different. In addition, the semantics remain different under perturbation (see Fig. 5 in the appendix). So we would expect the OOD detector to perform well in both standard and adversarial settings.
>
> - I agree that it would be helpful to have a plot showing how different are the distributions of in and out data in the feature space. However, we have tried to show this in Fig. 1, where we plot the detection score against the perturbation budget. So one can get a sense of how different are in and out data under different perturbation budgets.
>
> Please let me know if these have not addressed your concerns.

---

### Meta-Review · Area_Chair_Hk6z · 2022-08-26

**Recommendation:** Accept
**Confidence:** Certain

**Metareview:**

The paper considers OOD detection facing adversarial attacks. It shows existing defenses are ineffective to end-to-end attacks on both in/out distribution data with larger perturbation than previous work. With the intuition of training with adversarial examples on both in/out distribution data, it then proposed a method (ATD) that uses a generator model to create OOD samples. It provided experiments showing the method outperforms previous methods.

The paper considers an important topic and made significant contributions: new evaluation of existing methods under stronger attacks, a well-designed method, quite thorough experiments and strong performance of the proposed method.

The reviewers have some concerns but are largely addressed by the authors' repsonses. The major ones are:

1. Missing citation of/comparison to some existing work. Though ATOM only considers out distribution robustness, it is suggested that the authors include the comparison in the revision for completeness. In the response, the authors have provided results for the comparison. There are also related certified defenses [Bitterwolf20, Meinke21]. The response also added discussion about these.

2. Details should be included in the appendix, e.g., detailed experimental results, evaluation details like hyperparameters, etc. The revision has added these details.

3. Evaluation on more combinations of the ID / OOD datasets. The authors have also provided more in the response; it is suggested to also include them in the revision.

Overall, the authors have made significant improvements addressing the reviewers' concerns/suggestions. The revised paper is of good quality for acceptance.


**Award:**

No

---

### Decision · Program_Chairs · 2022-09-14

Accept